# Human pathogenic bacteria on high-touch dry surfaces can be controlled by warming to human-skin temperature under moderate humidity

Ayano Konno[1], Torahiko Okubo[1], Yoshiaki Enoeda[1], Tomoko Uno[2,3], Toyotaka Sato[4,5,6,7], Shin-ichi Yokota[4], Rika Yano[3], Hiroyuki Yamaguchi[1]*

1 Faculty of Health Sciences, Department of Medical Laboratory Science, Hokkaido University, Kita-ku, Sapporo, Japan, 2 Department of Nursing, Sapporo Medical University School of Medicine, Chuo-ku, Sapporo, Japan, 3 Faculty of Health Sciences, Department of Fundamental Nursing, Hokkaido University, Kita-ku, Sapporo, Japan, 4 Department of Microbiology, Sapporo Medical University School of Medicine, Chuo-ku, Sapporo, Japan, 5 Faculty of Veterinary Medicine, Laboratory of Veterinary Hygiene, Hokkaido University, Kita-ku, Sapporo, Japan, 6 Graduate School of Infectious Diseases, Hokkaido University, Kita-ku, Sapporo, Japan, 7 One Health Research Center, Hokkaido University, Kita-ku, Sapporo, Japan

* hiroyuki@med.hokudai.ac.jp

**Data Availability Statement:** All relevant data are within the paper and its Supporting Information files.

## Abstract

Healthcare-associated infections have become a major health issue worldwide. One route of transmission of pathogenic bacteria is through contact with "high-touch" dry surfaces, such as handrails. Regular cleaning of surfaces with disinfectant chemicals is insufficient against pathogenic bacteria and alternative control methods are therefore required. We previously showed that warming to human-skin temperature affected the survival of pathogenic bacteria on dry surfaces, but humidity was not considered in that study. Here, we investigated environmental factors that affect the number of live bacteria on dry surfaces in hospitals by principal component analysis of previously-collected data ($n = 576$, for CFU counts), and experimentally verified the effect of warming to human-skin temperature on the survival of pathogenic bacteria on dry surfaces under humidity control. The results revealed that PCA divided hospital dry surfaces into four groups (Group 1~4) and hospital dry surfaces at low temperature and low humidity (Group 3) had much higher bacterial counts as compared to the others (Group 1 and 4) ($p<0.05$). Experimentally, warming to human-skin temperature (37˚C with 90% humidity) for 18~72h significantly suppressed the survival of pathogenic bacteria on dry surfaces, such as plastic surfaces [$p<0.05$ vs. 15˚C (*Escherichia coli* DH5α, *Staphylococcus aureus*, *Pseudomonas aeruginosa*, *Acinetobacter baumannii*, and *bla*$_{NDM-5}$ *E. coli*)] or handrails [$p<0.05$ vs. 15~25˚C (*E. coli* DH5α, *S. aureus*, *P. aeruginosa*, *A. baumannii*)], under moderate 55% humidity. Furthermore, intermittent heating to human-skin temperature reduced the survival of spore-forming bacteria (*Bacillus subtilis*) ($p<0.01$ vs. continuous heating to human-skin temperature). NhaA, an Na$^+$/H$^+$ antiporter, was found to regulate the survival of bacteria on dry surfaces, and the inhibitor 2-aminoperimidine enhanced the effect of warming at human-skin temperature on the survival of pathogenic bacteria (*E. coli* DH5α, *S. aureus*, *A. baumannii*) on dry surfaces. Thus, warming to human-

**Funding:** This study was funded by a Grant-in-Aid for Scientific Research, KAKENHI (grant number: 20K20613) (to H.Y.). The funders had no role in the study design, data collection and analysis, decision to publish, or preparation of the manuscript.

**Competing interests:** The authors have declared that no competing interests exist.

**Abbreviations:** HAIs, healthcare-associated infections; CFUs, colony-forming units; PCA, principal component analysis; PCs, principal components; NhaA, Na$^+$/H$^+$ antiporter A; 2-AP, 2-aminoperimidine.

skin temperature under moderate humidity is a useful method for impairing live pathogenic bacteria on high-touch surfaces, thereby helping to prevent the spread of healthcare-associated infections.

## Introduction

The occurrence and spread of healthcare-associated infections (HAIs) caused by human pathogenic bacteria in public medical facilities, including long-term care homes and nursing homes, has become a global problem [1–3]. Compared with community-acquired infections in the general population, HAIs often affect patients who are older and have weakened immune systems, making them more difficult to treat and often negatively impacting on the quality of life and prognosis of patients [4]. The World Health Organization reported that the spread of HAIs prolonged hospital stays, increasing the financial burden on healthcare systems and medical insurance premiums, as well as increasing the risk of emergence of resistant bacteria [5, 6]. For these reasons, there is an urgent need for new strategies to prevent HAIs.

One factor contributing to the spread of HAIs is contact between medical staff and patients, in particular, through frequently-touched dry surfaces, such as handrails and doorknobs, referred to as "high-touch surfaces" [7]. Such surfaces become hot spots for infection because human pathogenic bacteria adhere to them and survive for a long period of time [7]. Therefore, at present, as a preventive measure against HAIs, regular cleaning of dry surfaces using disinfectants is performed at hospitals. However, such cleaning procedures are costly and do not prevent outbreaks of microorganisms that are resistant to disinfectant. In fact, it has been reported that extended spectrum β-lactamase-producing *Escherichia coli* strains are resistant to multiple disinfectants [8] and *Acinetobacter* possesses a chlorhexidine efflux pump [9]. Furthermore, *Pseudomonas aeruginosa*, a bacterium that causes opportunistic infections, reportedly acquires resistance to multiple antibiotics as a result of the persistence of disinfectants containing benzalkonium chloride on regularly cleaned surfaces [10]. It has also been reported that, despite routine cleaning for the management of HAIs, vancomycin-resistant enterococci and methicillin-resistant *Staphylococcus aureus* persist in the healthcare environment for days, threatening the spread of infection [11]. This is partly due to the overuse of antibiotics or disinfectants [12].

Biofilms formed on dry surfaces cannot be permeated by disinfectant, and bacteria continue to survive and cause a high frequency of infections in hospitals after cleaning [13]. Several studies have shown that the cleanliness of contact surfaces is inadequate in terms of preventing infection [14, 15]. New control measures therefore need to be developed to replace regular cleaning with disinfectants.

Dry surfaces are an unsuitable environment for bacteria to maintain vital metabolic activity [16], and bacteria have therefore developed strategies to resist drying. However, the survival mechanisms that bacteria employ in dry environments are not fully understood. Environmental factors such as temperature and humidity have been reported to contribute to the survival of bacteria in dry environments [16, 17]. In our previous studies, we clarified that temperature affected the population of airborne bacteria in an underground pedestrian space in Sapporo, Japan [18], as well as the number of live bacteria on dry surfaces in a hospital environment [19, 20]. Furthermore, we reported preliminary evidence that the survival of pathogenic bacteria was significantly impaired on dry-plastic surfaces warmed to 37˚C [21]. These findings strongly suggest that the number of live bacteria can be controlled on dry surfaces simply by adjusting the temperature.

Here, we extend these findings by investigating the dynamics of bacteria on dry surfaces in hospitals depending on changes in temperature and humidity, and the degree of human

contact. We show experimentally that human pathogenic bacteria on dry surfaces such as handrails can be controlled by warming of the surfaces to human-skin temperature under moderate humidity. Furthermore, we demonstrate that NhaA, an $Na^+/H^+$ antiporter, is a crucial factor in supporting the survival of bacteria under dry conditions, and that an inhibitor (2-aminoperimidine: 2-AP) of NhaA enhances the suppressive effects of warming.

## Results

### Low temperature and humidity increase the survival of bacteria on high-touch dry surfaces in hospital environments

To clarify the relationship between the number of live bacteria on hospital surfaces of different wards and environmental factors (temperature, humidity, and human contact), the data (66 sets) published in our previous manuscript were reused [19] and subjected to principal component analysis (PCA). The data set consisted of the number of colony-forming units (CFUs) on the high-touch surfaces of different wards [Obstetrics (O), Surgery, (S), or Internal medicine (I)] and environmental factors [temperature (T1–3), humidity (M1–3), and the number of human contacts (N1–3)] in three hospitals of different sizes (designated H, K, and M) located in Sapporo, Japan. PCA revealed a cumulative contribution rate, which was the sum of the first (44.1%) and second (31.4%) principal components (PCs), of 75.5%, and the main composite variables of PC1 and PC2 were humidity (negative impact) and temperature (negative impact), respectively (Fig 1A). Based on these variables, a scatter plot was created with four groups, consisting of low temperature and high humidity in Group 1, high temperature and high humidity in Group 2, low temperature and low humidity in Group 3, and high temperature and low humidity in Group 4 (Fig 1A). The groupings were significantly different between the hospitals, which would likely affect the number of viable bacteria on dry surfaces. As expected, the number of live bacteria on the dry surfaces of Group 3, which tended to be low in temperature and humidity, was significantly higher than for the other groups (Fig 1B and S1 Fig). The ATP level was used as an index for the frequency of human contact, but there was no difference between the groups, indicating that differences in human contact frequency did not affect live-bacterial counts on dry surfaces (Fig 1C and S2 Fig). A reduction in humidity and temperature promoted the survival of environmental bacteria on high-touch surfaces, and conversely, an increase in temperature to human-skin temperature, as we previously reported [21], decreased the survival of bacteria on dry surfaces, potentially preventing the spread of HAIs.

### Warming to human-skin temperature with high humidity significantly reduces the survival of human pathogenic bacteria on dry surfaces

We previously found that the number of live bacteria on dry surfaces in hospitals was dramatically influenced by a change in temperature, along with some other contributing environmental factors [19, 20]. We reported preliminary evidence that the survival of pathogenic bacteria was significantly impaired on dry-plastic surfaces warmed to 37°C [21]. In the current study, we employed PCA to investigate the effect of temperature and humidity on the survival of *E. coli* DH5α, and other human pathogenic bacteria, on dry surfaces. First, we determined the survival rate of *E. coli* on a dry plastic surface in an incubator adjusting the humidity from 45%–90% and the temperature from 25°C–37°C, and then estimated the most effective conditions using PCA. The results revealed that the cumulative contribution rate, which was the sum of the first (53.8%) and second (33.3%) PCs, was 87.1%, and the main composite variables of PC1 and PC2 were temperature (negative impact) plus survival rate (positive impact) and humidity (positive impact), respectively. The most significant reduction in the survival rate of

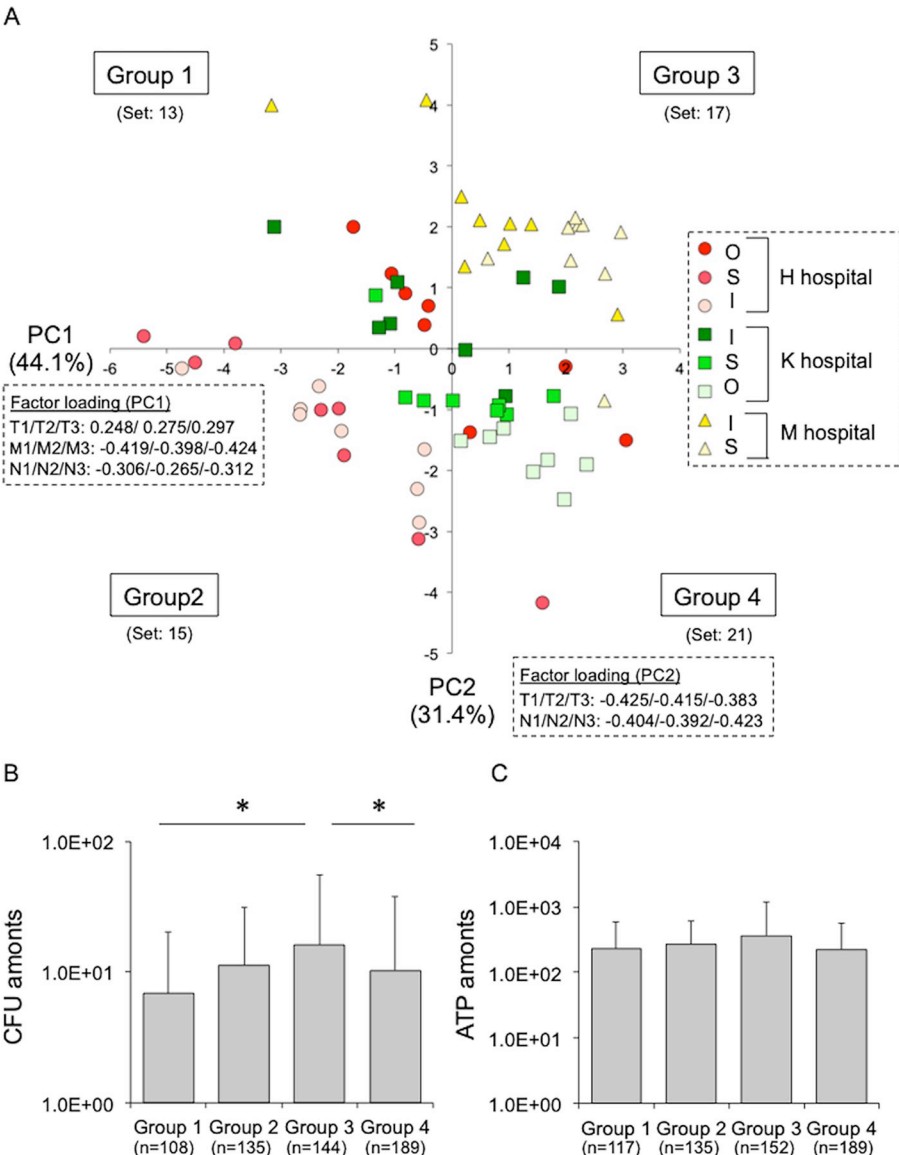

**Fig 1. Low temperature and humidity in hospitals increases the survival rate of bacteria on high-touch dry surfaces. A.** PCA scatter plot showing four groups (Groups 1–4) with distinct environmental conditions [temperature (T: T1–3), humidity (M: M1–3), and number of people (N: N1–3)] in three hospitals [designated H, K, and M]. PC1-factor loading shows 0.248(T1)/ 0.275(T2)/ 0.297(T3), -0.419(M1)/ -0.398(M2)/ -0.424(M3), and -0.306(N1)/ -0.265(N2)/ -0.312(N3) (N: number of people at that time). PC2-factor loading shows -0.425(T1)/ -0.415(T2)/ -0.383 (T3) and -0.404(N1)/ -0.392(N2)/ -0.423(N3). As specified in the text, the data (66 sets) published in our previous manuscript were reused [19]. "O" ward, Obstetrics. "S" ward, Surgery. "I" ward, Internal medicine. "%," cumulative contribution rates for PC1 and PC2. See S1 Table. **B.** Comparison of the number of live bacteria (CFU) on dry surfaces between Groups 1–4. Bars [Group 1 (*n* = 108), Group 2 (*n* = 135), Group 3 (*n* = 144), and Group 4 (*n* = 189)] show the average ± SD. *, *p*<0.05, with a statistically significant difference between the combinations connected by a line. **C.** Comparison of the amount of ATP (as an index for the frequency of "human contact") on dry surfaces between Groups 1–4. Bars [Group 1 (*n* = 117), Group 2 (*n* = 135), Group 3 (*n* = 157), and Group 4 (*n* = 189)] show the average ± SD.

*E. coli* was observed at 37°C and 90% humidity, and the least significant effect was observed at 25°C and 45% humidity (S3 Fig). In particular, increasing the temperature to 37°C had a gradual effect on the survival rate, but this effect was considerably diminished at humidity below

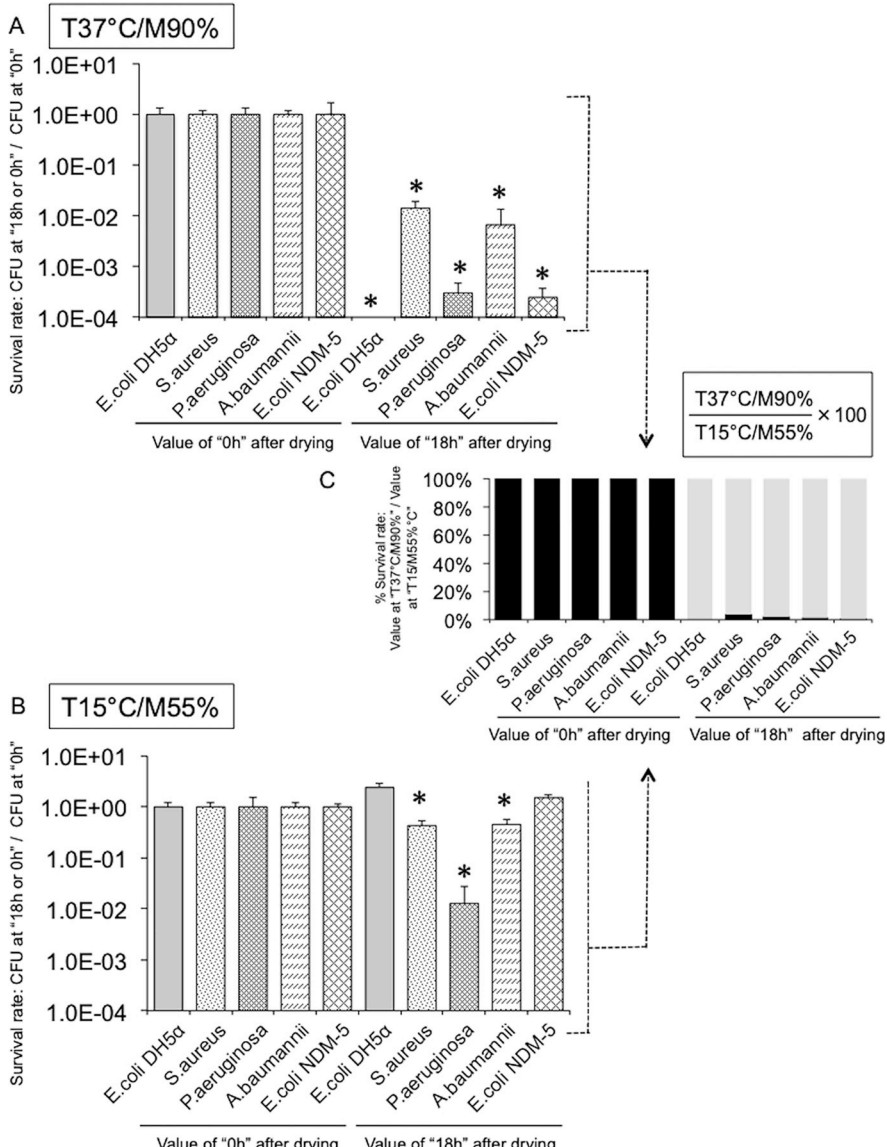

**Fig 2. The warming of dry surfaces to human-skin temperature affects the survival of human pathogenic bacteria** [*S. aureus*, *P. aeruginosa*, *A. baumannii*, and *bla*$_{NDM-5}$ *E. coli* (NDM-5)] as well as *E. coli* DH5α. The experiment was performed in a thermo-hygrostat incubator. Temperature (T) (15°C and 37°C) and humidity (M) (55% and 90%) were adjusted by the controller installed in the incubator. See the Materials and Methods. **A.** Survival rate of the bacteria under the most effective conditions (T 37°C/ M 90%). "Survival rate" shows the ratio of CFUs at "18 h" / "0 h" after drying. Experiments were performed at least three times. Bars show the average ± SD. *, $p < 0.05$, with a statistically significant difference compared with immediately after drying ("0 h"). **B.** Survival rate of the bacteria under the least effective conditions (T 15°C/M 55%). See above. **C.** Survival rates (%) of the bacteria between the most effective [T 37°C/M 90% (A)] and the least effective [T 15°C/M 55% (B)] conditions.

60% (S4 Fig), even with warming to human-skin temperature. Furthermore, the warming effect was similarly observed for other human pathogenic bacteria (*S. aureus*, *P. aeruginosa*, *Acinetobacter baumannii*, and *bla*$_{NDM-5}$ *E. coli)* as well as *E. coli* DH5α (Fig 2). However, warming to 37°C had no effect on the spores of *Bacillus subtilis*, but intermittent warming at human-skin temperature significantly reduced the number of live bacteria (S5 Fig). Thus,

warming to human-skin temperature along with increasing humidity remarkably suppressed the survival of human pathogenic bacteria on dry surfaces.

## Warming of handrail surfaces to human-skin temperature with moderate humidity significantly impairs the survival of human pathogenic bacteria

Finally, we verified the effect of direct heating of a dry handrail surface to human-skin temperature under moderate humidity (55%) on the survival of human pathogenic bacteria. Warming to human-skin temperature was performed using a detachable motorcycle hand heater, and a commercially-available stainless handrail designed for doors was used as an experimental device (Fig 3A, left). The handrail (at the point closest to the heater) was warmed to human-skin temperature (approximately 37°C), with the temperature gradually decreasing depending on the distance from the heater (Fig 3A, right). As expected, the survival rate of *E. coli* DH5α decreased significantly on the handrail surface warmed to human-skin temperature (Fig 3B). The effect became more pronounced with time (Fig 3C). To confirm this killing effect, we stained live/dead bacteria recovered from each point on the heated handrail. Similar to the CFU data, the relative staining of *E. coli* DH5α significantly decreased with proximity to the heating zone (S6 Fig). Furthermore, the inhibitory effect of heating was confirmed for other human pathogenic bacteria (*S. aureus*, *P. aeruginosa*, and *A. baumannii*), but was least effective against *S. aureus* (Fig 4A). We next identified a dry-resistance factor (NhaA: Na$^+$/H$^+$ antiporter) from the Tn-mutant library of *E. coli* DH5α ($n = 243$) (S7 Fig), and examined the effect of treatment with an NhaA inhibitor (2-aminoperimidine, 2-AP) (Fig 4B). As expected, 2-AP significantly enhanced the effect of warming to human-skin temperature on the survival of *S. aureus* and *A. baumannii* on dry surfaces, both of which are naturally resistant to drying [9–12, 22, 23] (Fig 4C). It has also been confirmed that 2-AP is effective against *E. coli* DH5α (data not shown). Thus, warming to human-skin temperature under conditions of moderate humidity remarkably impaired the survival rate of human pathogenic bacteria on dry surfaces.

## Discussion

Here, we show that environmental temperature and humidity have a great impact on the survival of bacteria on high-touch dry surfaces. In particular, hospital surfaces at low temperature and humidity tended to harbor more live bacteria. It is however impossible to increase the humidity of hospital environments beyond a specific percentage as this might lead to other negative effects, for example leading to the growth of mold. Furthermore, we experimentally demonstrated that pathogenic bacteria applied to dry surfaces were quickly killed by warming to human-skin temperature with moderate humidity. Thus, we succeeded in controlling the survival of pathogenic bacteria on dry surfaces simply by warming to human-skin temperature without relying on disinfectants. This has potential implications for preventing the transmission of pathogenic bacteria.

It is recommended by the American Society of Heating, Refrigerating and Air-conditioning Engineers Standard that the temperature in hospitals should be controlled between 20°C–27°C with 40%–60% humidity for comfort [24]. The indoor temperature and humidity varied in the hospitals that we assessed in Japan. For example, the temperature and humidity in hospital "M" were set lower than those in the other hospitals tested, and more live bacteria were detected. Because there was no difference in the amount of ATP (an index of human contact) on the dry surfaces in this hospital, the higher number of live bacteria was not a result of a higher frequency of human contact. Thus, the temperature and humidity arbitrarily set by the air conditioning system affected the microenvironment of the dry surfaces, directly influencing the survival of live bacteria on these surfaces. This confirmed that a low temperature and low humidity are advantageous to the survival of pathogenic bacteria on dry surfaces.

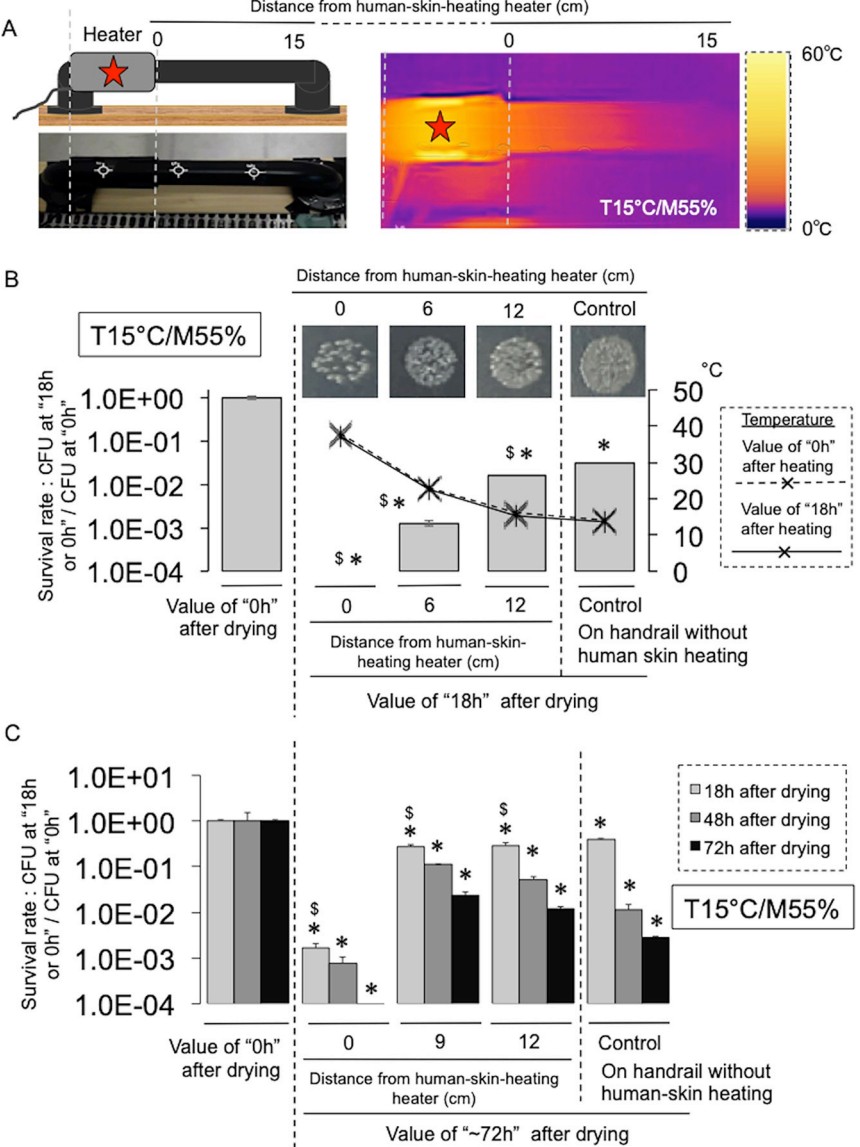

**Fig 3. Heating of handrail surface to human-skin temperature with moderate humidity significantly impairs the survival of *E. coli* DH5α.** The experiment was performed in a thermo-hygrostat incubator. Temperature (T) (15˚C) and humidity (M) (55%) were adjusted by the controller installed in the incubator. See the Materials and Methods. **A.** Handrail device and heatmap showing the temperature distribution across the handrail device when connected to the heater. Warming was performed to conditions of 15˚C and 55% humidity (T 15˚C/M 55%). Red star shows the location of the heater on the handrail. **B.** Survival rate of *E. coli* DH5α at each measurement location. The experiment was performed at least three times. Values show the ratio of CFUs between immediately after drying "0 h" and "18 h" after drying. Images show representative colonies of smear samples on LB agar plates collected from the handrail. Dashed and solid lines show the changing temperature due to heating. Bars show the average ± SD. *, $p<0.05$, with a statistically significant difference compared with the value immediately after drying "0 h." $, $p<0.05$, with a statistically significant difference compared with the "Control (without heating)" value. **C.** Effect of handrail warming to human-skin temperature on the survival of *E. coli* DH5α over 72 h. See above. Bar shows the average ± SD. *, $p<0.05$, with a statistically significant difference compared with the value immediately after drying "0 h" for each time course. $, $p<0.05$, with a statistically significant difference compared with the "Control (without heating)" value for each time course.

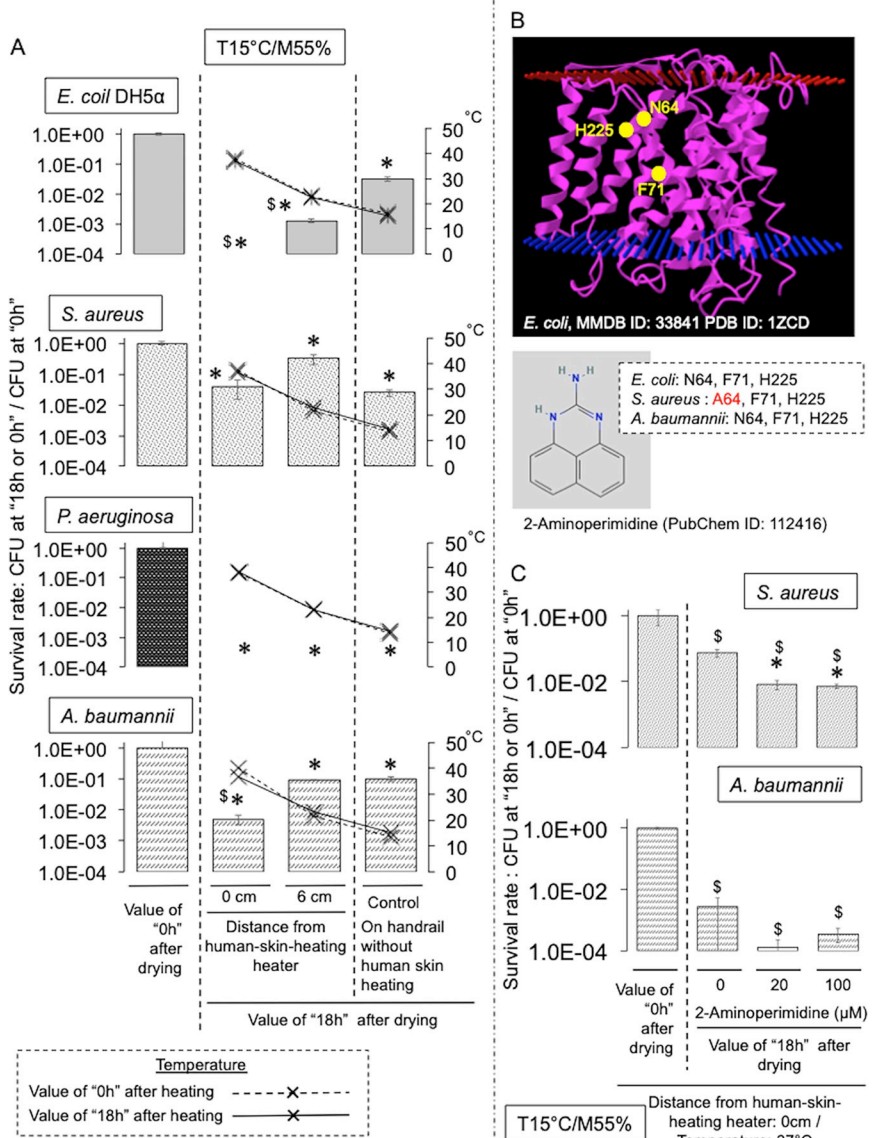

**Fig 4. Heating of the handrail surface to human-skin temperature with moderate humidity significantly impairs the survival of other pathogenic bacteria (*S. aureus*, *P. aeruginosa*, *A. baumannii*) as well as *E. coli* DH5α, and an inhibitor (2-AP) of NhaA, which regulates dry resistance, enhances the warming effect against some bacteria.** The experiment was performed in a thermo-hygrostat incubator. Temperature (T) (15°C) and humidity (M) (55%) were adjusted by the controller installed in the incubator. See the Materials and Methods. **A.** Survival rates of human pathogens (*S. aureus*, *P. aeruginosa*, *A. baumannii*) and *E. coli* DH5α on the handrail device warmed to human-skin temperature. Warming was performed under conditions of 15°C and 55% humidity (T 15°C/M 55%). Values show the ratio of CFUs between immediately after drying "0 h" and "18 h" after drying. Bars show the average ± SD. *, $p < 0.05$, with a statistically significant difference compared with the value immediately after drying "0 h." $, $p < 0.05$, with a statistically significant difference compared with the "Control (without heating)" value. **B.** 3D structures of annotated protein sequences of *E. coli* NhaA (MMDB ID: 33841 PDB ID: 1ZCD) acquired from the NCBI Structure library (https://www.ncbi.nlm.nih.gov/), and the chemical structure of the specific inhibitor (2-aminoperimidine, 2-AP: PubChem ID: 112416) obtained from the PubChem Compound library at the same site. Yellow marks indicate binding sites for 2-AP. Among the pathogens used in the experiment, only *S. aureus* replaced "N64" with "A64." Red ladder, extracellular membrane. Blue ladder, inner membrane. **C.** 2-AP enhances the warming effect on some bacteria (*S. aureus* and *A. baumannii*). See above. Bars show the average ± SD. $, $p < 0.05$, with a statistically significant difference when compared with the value immediately after drying "0 h." *, $p < 0.05$, with a statistically significant difference when compared with those without 2-AP at "18 h after drying".

We performed matrix analysis (temperature range: 25˚C–37˚C, humidity: 45%–90%) on plastic plates to determine the optimal temperature and humidity that could control live pathogenic bacteria on dry surfaces. As expected, in contrast to a low temperature with low humidity, warming to human-skin temperature (37˚C) with increasing humidity (~90%) dramatically impaired the survival rates of other pathogenic bacteria as well as *E. coli* DH5α. Two pathogenic bacteria (*S. aureus* and *A. baumannii*), which are known to exhibit resistance to desiccation under dry conditions as a result of a thick cell wall or effective trehalose accumulation [25–27], were also susceptible to heating and humidity, but to a lesser extent than *E. coli*. Heating to human-skin temperature had no effect on spore-forming bacteria (*B. subtilis*), but intermittent warming (every 2 h) to human-skin temperature significantly reduced (to one-tenth) the number of live bacteria on dry surfaces. This suggested that warming has a potential inhibitory effect on spore-forming *Clostridioides difficile*, which diffuses diarrhea in the setting of recent healthcare [28, 29]. Thus, warming to human-skin temperature with moderate humidity is likely to have a general suppressive effect on pathogenic bacteria on dry surfaces.

As expected, when a stainless steel handrail device was directly warmed to human-skin temperature using a motorcycle hand heater, the survival rates of pathogenic bacteria significantly decreased. It is reported that bacteria may enter a dormant state stochastically in microbial populations in response to changes in culture conditions such as exposure to antibiotics or heating [30, 31]. However, in our study, the results of live/dead staining confirmed that the pathogenic bacteria had not entered a dormant state and that this did not explain the decrease in cultured cells from the dry surfaces. We used a stainless steel handrail in our experiments, which is a material that is widely available and widely used in public spaces, but it is worth noting that other materials may vary in their ability to conduct heat and therefore the effects of heating may be weakened. In fact, our previous study showed that the survival rate of bacteria differed depending on the material used for the dry surfaces in a hospital environment [32]. Further studies should address this issue by testing materials in terms of the effect of warming to human-skin temperature.

At present, the detailed mechanism by which heating weakens the survival rate of pathogenic bacteria on dry surfaces remains unknown. However, accumulating evidence indicates that temperature affects the survival of pathogenic bacteria on several surfaces by drying. For example, *Salmonella enterica* on an eggshell surface was found to grow under low-temperature (4˚C–12˚C) with low-humidity (43%) rather than high-temperature (25˚C) with high-humidity (85%) [33]. It has also been reported that *S. enterica* or enterohemorrhagic *E. coli* on a plastic tray died more rapidly under hot (25˚C) rather than cold conditions (5˚C) [34]. Furthermore, it has been reported that pathogens [*E. coli* O157:H7, *Shigella sonnei*, *S. enterica*, *Clostridium perfringens*, hepatitis A virus (HAV), feline calicivirus (FCV), and coliphage PRD1] were killed significantly more effectively under higher humidity conditions [35]. Taken together, these studies support our finding that warming to human-skin temperature can control the survival of pathogenic bacteria on dry surfaces.

We newly identified NhaA as a dry-resistance factor by screening a Tn-insertion mutant library of *E. coli* DH5α. The NhaA transporter is reported to be important for regulating and maintaining the pH in and around bacteria [36], and is extremely well conserved from bacteria to eukaryotes [37]. Although the role of NhaA in drying is not fully understood, it is likely responsible for controlling the leakage of ions from bacterial cells whilst maintaining bacterial structures [38, 39]. A hydrophobic NhaA inhibitor, 2-AP, readily penetrates the bacterial cell wall and binds specifically to three amino acids of NhaA (binding residues for *E. coli*: N64, F71, H225), inhibiting its transport activity [38, 39]. As expected, a synergistic effect was observed between warming to human-skin temperature and 2-AP for a pathogenic bacterium (*S. aureus*). In the case of staphylococci, which have thick cell walls, it may not be possible to completely disrupt the water balance in bacterial cells by warming to human-skin temperature

alone. However, there may be a potential application for 2-AP, which is inexpensive to produce, as a spray for the daily cleaning of dry surfaces in hospital environments.

In addition, alcohol (60–95%) is most commonly used to disinfect high-touch surfaces such as doorknobs and handrails, and its effectiveness with optimum bactericidal activity is well documented [40, 41]. However, since alcohol disinfection requires regular and continuous implementation, it is also true that regular cleaning that mainly use alcohol are time-consuming, manpower-intensive, and costly [42]. Therefore, if high-touch surfaces can be treated with 2-AP as well as keeping human-skin temperature, it would be possible to save time and effort in cleaning work, which may contribute to improving the quality of medical care by reducing work.

Meanwhile, the control of HAIs confers a significant burden on resources in healthcare settings [43]. Traditional cleaning methods are insufficient for decontamination, and other approaches have been proposed, including new disinfectants, steam, automated dispersal systems, and antimicrobial surfaces [11]. Because disinfection using chemical substances raises concerns regarding the emergence of resistance, new treatment strategies are needed [44]. The warming of high-touch dry surfaces may represent a new method of controlling the transmission of pathogenic bacteria in hospitals. At this point, it could not deny that some bacteria may also develop resistance to this slight increase in the temperature, responsible for developing resistance to warming on dry surfaces.

## Conclusions

The dry surfaces that our hands come into contact with are crucial mediators of infectious disease transmission via pathogenic bacteria. However, there is currently no reliable method for controlling live pathogenic bacteria on "high-touch" surfaces. In hospitals, high-touch surfaces are regularly cleaned with disinfectant, but this does not prevent nosocomial infections. Here, we found that the survival of pathogenic bacteria can be controlled on dry surfaces simply by warming to human-skin temperature under conditions of moderate humidity. Furthermore, NhaA (Na$^+$/H$^+$ antiporter) plays a role in regulating the survival of bacteria on dry surfaces, and the inhibitor 2-AP may have potential applications in controlling pathogenic bacteria on dry surfaces in hospital environments. Our findings may contribute toward creating safer public environments with a decreased risk of infectious disease transmission by employing a novel method that does not rely on chemicals such as disinfectants.

## Materials and methods

### Data set reused from our previous study

The data (66 sets) relating to the "number of people," "temperature," "humidity," "CFU numbers," and "ATP amounts" were reused from our previously published research [19] and subjected to PCA or a multiple comparison test (S1 Table). The data sets were collected from three hospitals [at different wards: Internal Medicine (I), Surgery (S), or Obstetrics and Gynecology (O)] of different sizes located in Japan (Sapporo city) from October 2011 to February 2012 [hospital "H" (>500 beds), hospital "K" (100–500 beds), and hospital "M" (<100 beds)].

### Bacterial strains used for this study

The following bacteria were used in this study: *E. coli* (DH5α), *P. aeruginosa* (ATCC27853), *A. baumannii* (ATCC19606), *bla*$_{NDM-5}$ *E. coli* (clinical isolate), *S. aureus* (ATCC29213), and spore-forming *B. subtilis* (laboratory stock). The bacteria were grown in Luria–Bertani (LB) medium (Nacalai Tesque, Kyoto, Japan) or nutrient medium (Nissui Pharma, Tokyo, Japan).

### Evaluation of the survival of pathogenic bacteria on dry surfaces under controlled temperature and humidity

To determine the most effective conditions for inhibiting the survival of *E. coli* DH5α on the dry surface of a 96-well plastic plate, various combinations of temperature (25˚C–37˚C) and humidity (45%–90%) were evaluated using a thermo-hygrostat incubator [IW223 (Yamato Scientific Co., Ltd., Tokyo, Japan) or KCL-2000W (EYELA, Tokyo Rikakikai Co., Ltd., Tokyo, Japan)]. In brief, the bacteria were suspended in LB liquid medium to $1.0×10^{10}$ CFU/mL, and 5 µL of solution was spotted onto a 96-well plate (U-type). After drying completely (T0), the plate was incubated in the thermo-hygrostat incubator for 18 h (T18). The dried bacteria in the wells were suspended in 100 µL of LB liquid medium and the number of live bacteria was counted. The live bacterial count was expressed as the ratio of the decreased value at "T18" to the value at "T0." The data sets for "temperature," "humidity," and "the survival rate of *E. coli*" were used for PCA or a multiple comparison test (S2 Table). Other pathogenic bacteria (*P. aeruginosa* ATCC27853, *A. baumannii* ATCC19606, $bla_{NDM-5}$ *E. coli*, and *S. aureus* ATCC29213) were similarly verified under the most and least effective conditions for the survival of *E. coli* DH5α on the dry surface. In addition, the effect of intermittent warming to human-skin temperature on spore-forming bacteria (*B. subtilis*) was verified (see the protocol in S5A Fig).

### Evaluation of the survival of human pathogenic bacteria on handrail surfaces warmed to human-skin temperature with moderate humidity

Commercially available stainless steel (length: 34 cm, diameter: 3.3 cm) placed sideways on wooden boards was used to simulate a "handrail dry surface" with a 9.2 cm-long-grip heater (OPMID, Osaka, Japan) attached to one end of the steel. This handrail device was placed into the thermo-hygrostat incubator adjusted to 15˚C (temperature) and 55% (humidity). There were two adjacent handrails; the handrail temperature was monitored by thermography (FLIR ONE Pro, Teledyne Flir, Wilsonville, OR, USA). One handrail was used as a control without heating, and the other was heated with a heater as below. The output of the heater had three settings (5 W, 7.5 W, 10 W) and the 10 W setting was selected as this heated the adjacent handrail to approximately 37˚C when inside the incubator adjusted at 15˚C with 55% humidity. The number of live bacteria on the handrail was determined as follows. In brief, the bacteria were suspended in 0.85% saline to a concentration of $1.5×10^{10}$ CFU/mL. Each 10 µL of solution was dropped onto a release paper of double-sided tape cut into 1.5 cm squares ($1.5×10^{8}$ CFU/spot). In some experiments, 2-AP (~100 µM) was added to the solution. After drying, the bacteria-spotted tapes were applied at different distances [0 cm (right next to the heater)– 15 cm] from the heater on the handrail device, and then incubated in the thermo-hygrostat incubator adjusted at 15˚C (temperature) and 55% (humidity) for ~72 h (T18, T48, T72). Each tape was then collected, suspended in 500 µL of PBS, and the number of live bacteria was counted. The live bacterial count was expressed as the ratio of the decreased value at "T18–T72" to the value at "T0." The viability of *E. coli* on the handrail was also quantified by an image analyzer (Kyence, Osaka, Japan) after Live/Dead BacLight staining (Molecular Probes, Eugene, OR, USA).

### Screening of a mutant with the least dry resistance from a Tn-insertion *E. coli* DH5α library and the generation of strains in which the gene was disrupted/reintroduced

A Tn-insertion mutant library of *E. coli* DH5α was constructed using the EZ-Tn5™ <KAN-2>Tnp Transposome™ kit (AR Brown Co., Ltd., Tokyo, Japan), according to the manufacturer's protocol. The library consisted of 243 mutant strains. The bacterial solution (5 µL,

approximately $1.0 \times 10^5$ CFU/spot) was spotted onto a 96-well plastic plate (U type), and then dried at room temperature. The number of CFUs was compared between the immediately dried (T0) and 24-h incubated (T24) samples at room temperature (approximately 22°C), and the results were expressed as a ratio (T24/T0). The Tn-inserted gene of a mutant strain with the least dry resistance was identified by arbitrary primed PCR, according to a previous report [45]. The primers used for PCR were as follows: 1st PCR TnF: 5′-GCA ATG TAA CAT CAG AGA GAG ATT TTG AG-3′, 1st PCR R: 5′-GGC CAC GCG TCG ACT AGT ACN NNN NNN NNN GAT AT-3′, 2nd PCR TnF: 5′-AGC TTC AGG GTT GAG ATG TG-3′, 2nd PCR R: 5′-GGC CAC GCG TCG ACT AGT AC-3′. Amplified DNA was sequenced, and then genes were determined based on matches to the NCBI database (https://www.ncbi.nlm. nih.gov/). Because *nhaA* was the only site of Tn-insertion in the mutant strain with markedly reduced dry resistance, strains in which the gene was disrupted by recombination and reintroduced by complementation were established. In brief, recombination was performed using a kanamycin resistance cassette inserted into sites 50 bp upstream (*nhaA*-F 5′-GCG GGG TAA AAT AGT AAA AAC GAT CTA TTC ACC TGA AAG AGA AAT AAA AAA ATA ATT A-3′) and downstream (*nhaA*-R 5′-TGA TAA CAA TGA AAA GGG AGC CGT TTA TGG CTC CCC GGT AAA CCG TCC TGT AAT ACG AC-3′) of the *nhaA* gene in the cloning site of plasmid pKD46 encoding the recombinase. The complementary strain was established by transformation of pSTV28 (Clontech Takara Cellartis, Shiga, Japan) containing the *nhaA* gene.

## Three-dimensional (3D) structure of NhaA and the binding sites of the 2-aminoperimidine inhibitor

The 3D structures of annotated protein sequences of *E. coli* NhaA (MMDB ID: 33841, PDB ID: 1ZCD) were acquired from the NCBI structure library (https://www.ncbi.nlm.nih.gov/), and the chemical structure of the specific inhibitor (2-aminoperimidine: PubChem ID: 112416) was obtained from the PubChem compound library at the same site.

## PCA and statistical analysis

The two datasets [field study (temperature, humidity, and the number of people in three city hospitals in Sapporo) (See S1 Table), experimental study (temperature, humidity, and the survival rate of *E. coli*) (See S2 Table)] were separately compressed into two-component synthetic variables by PCA using R (Version 1.0.136), and the groups were visualized in Excel (Version 14.7.3). Multiple comparison tests were performed using the Bonferroni/Dunn method or the Steel–Dwass method. A *p*-value of less than 0.05 ($p < 0.05$) was considered significant.

## Supporting information

**S1 Table. Data sets (temperature, humidity, number of people, and the number of bacteria (CFU) / the frequency of human contact (ATP) in three city hospitals in Sapporo) reused from our previously published manuscript [19].** "CFU" indicates the number of live bacteria on dry surfaces in hospitals, "ATP" shows the human-contact frequency for these surfaces. (PDF)

**S2 Table. Data sets obtained from experimental matrix analysis (temperature, humidity, and the survival rate of *E. coli*).** "Survival rate" indicates the value (number of live *E. coli*) for "T18" vs "T0." (PDF)

**S1 Fig. Comparison of the number of live bacteria at each sampling site among the four groups classified by environmental factors.** Group 1 was low temperature and high humidity [total 108: $n = 12$ (each bar)], group 2 was high temperature and high humidity [total 135: $n = 15$ (each bar)], group 3 was low temperature and low humidity [total 144: $n = 16$ (each bar)], and group 4 was high temperature and low humidity [total 21: $n = 21$ (each bar)]. See the detailed sampling sites in S1 Table. Bars show the average ± SD. *, $p < 0.05$ with statistical significance between specific groups.
(PDF)

**S2 Fig. Comparison of the amount of ATP at each sampling site among the four groups classified by environmental factors.** Group 1 was low temperature and high humidity, [total 117: $n = 12$ (each bar)], group 2 was high temperature and high humidity [total 135: $n = 15$ (each bar)], group 3 was low temperature and low humidity [total 152: $n = 17$ (each bar excluding "locker, $n = 16$")], and group 4 was high temperature and low humidity [total 189: $n = 21$ (each bar)]. See the detailed sampling sites in S1 Table. Bars show the average ± SD. *, $p < 0.05$ with statistical significance between specific groups.
(PDF)

**S3 Fig. Warming to human-skin temperature with moderate humidity significantly reduces the survival rate of human pathogenic bacteria on dry surfaces of a 96-well plastic plate.** The experiment was performed in a thermo-hygrostat incubator. Various combinations of temperature (T) (25˚C -37˚C) and humidity (M) (45%-90%) were adjusted by the controller installed in the incubator. See the Materials and Methods. **A.** PCA scatter plot showing the impact on the survival rate of *E. coil* of temperature (25˚C–37˚C) and humidity (45%–90%). PC1-factor loading shows -0.646 (T: temperature), -0.288 (M: humidity), and 0.707 (S: the survival rate of *E. coli*). PC2-factor loading shows -0.408 (T) and 0.913 (M). Six runs were performed for each combination of matrices. **B.** Comparison of the survival rate of *E. coli* on dry surfaces (18 h/0 h) with 45%–90% humidity at 25˚C. Bars ($n = 6$) show the average ± SD. *, $p < 0.05$, with a statistically significant difference compared with the conditions for highest survival (T25/M45). **C.** Comparison of the survival rate of *E. coli* on dry surfaces (18 h/0 h) with 45%–90% humidity at 37˚C. Bars ($n = 6$) show the average ± SD. *, $p < 0.05$, with a statistically significant difference compared with the conditions for highest survival (T37/M45).
(PDF)

**S4 Fig. Detailed comparison of the survival rates of *E. coli* on dry surfaces (18 h/0 h) at 25˚C–37˚C and 45%–90% humidity using the data obtained from S3 Fig.** Bars ($n = 6$) show the average ± SD. *, $p < 0.05$, with a statistically significant difference compared with the conditions for highest survival for each panel.
(PDF)

**S5 Fig. Intermittent warming to human-skin temperature significantly reduced the number of spore-forming *B. subtilis*.** The experiment was performed in a thermo-hygrostat incubator. Cycle of temperature (22˚C and 37˚C) and humidity (RH: 45%-90%) were program-controlled by the controller installed in the incubator. See the Materials and Methods. **A.** Graphs show the schedule of intermittent warming and the temperature (left) and humidity (right) [blue: -B- (repeating 22˚C with 45% humidity and 37˚C with 90% humidity with an interval of 2 h)]. Fixed schedules were included as controls [red: -A- (22˚C and 45%), green: -C- (37˚C and 90%)]. **B.** Temperature and humidity monitoring with a data logger (AD-6324SET, AD Discover Precision, Tokyo, Japan). Red, temperature. Blue, humidity. Arrows indicate the timing of removal of the plate from the incubator. **C.** Effect of intermittent warming at human-skin temperature on decreasing the number of *B. subtilis* compared with *E. coli*.

\*\*, *p*<0.01, with a statistically significant difference compared with those of "red bar A."
(PDF)

**S6 Fig. Validity of the results that heating of handrails to human-skin temperature kills *E. coli* DH5α, evidenced by live/dead staining.** The experiment was performed in a thermo-hygrostat incubator with temperature 15˚C and humidity 55%. Images show representative staining patterns (× 1,000). Area (red or green) was measured as the number of pixels by ImageJ software. Survival rate is shown as a ratio (T24/T0). Experiments were performed at least three times. Bars show the average ± SD. \*, *p*<0.05, with a statistically significant difference compared with "cont."
(PDF)

**S7 Fig. Survival rate of Tn-inserted *E. coli* mutants [library with 243 mutants (red numbers)] on dry surfaces, and confirmation of its efficacy using *nhaA*-disrupted and -reintroduced strains.** The experiment was performed at room temperature (approximately 22˚C). **A.** The number of CFUs was compared between immediately-dried (T0) and 24-h incubated (T24) samples, and the results were expressed as a ratio (T24/T0). Red arrow indicates the mutant (no. 75) with the lowest resistance to desiccation among the library. **B.** Loss of dry resistance in the *nhaA*-disrupted strain and recovery in the reintroduced strain. The experiment was performed at least three times. Bars show the average ± SD. \*, *p*<0.05, with a statistically significant difference compared with "*E. coli* DH5α" for each day. $, *p*<0.05, with a statistically significant difference compared with "*E. coli* DH5αΔ*nhaA*::*nhaA*" for each day.
(PDF)

## Acknowledgments

We thank Mr Kazuki Morinaga and Ms Nao Fujii for their help with the laboratory research. A professional proofreading company, Edanz (https://jp.edanz.com/ac), edited a draft of this manuscript.

## Author Contributions

**Conceptualization:** Hiroyuki Yamaguchi.

**Data curation:** Torahiko Okubo, Tomoko Uno, Rika Yano, Hiroyuki Yamaguchi.

**Formal analysis:** Ayano Konno, Torahiko Okubo, Yoshiaki Enoeda, Tomoko Uno, Rika Yano.

**Funding acquisition:** Hiroyuki Yamaguchi.

**Investigation:** Hiroyuki Yamaguchi.

**Methodology:** Ayano Konno, Toyotaka Sato, Shin-ichi Yokota.

**Resources:** Toyotaka Sato.

**Software:** Tomoko Uno, Hiroyuki Yamaguchi.

**Supervision:** Torahiko Okubo, Shin-ichi Yokota, Rika Yano, Hiroyuki Yamaguchi.

**Validation:** Torahiko Okubo, Tomoko Uno, Rika Yano, Hiroyuki Yamaguchi.

**Visualization:** Tomoko Uno.

**Writing – original draft:** Hiroyuki Yamaguchi.

**Writing – review & editing:** Hiroyuki Yamaguchi.

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
