## [Decision Letter · Decision Letter 0]

22 Jun 2023

PONE-D-23-08650Human pathogenic bacteria on high-touched dry surfaces can be controlled by warming to human-skin temperature under moderate humidityPLOS ONE

Dear Dr. Yamaguchi,

Thank you for submitting your manuscript to PLOS ONE. After careful consideration, we feel that it has merit but does not fully meet PLOS ONE’s publication criteria as it currently stands. Therefore, we invite you to submit a revised version of the manuscript that addresses the points raised during the review process.

ACADEMIC EDITOR:Revision Requested

We look forward to receiving your revised manuscript.

Kind regards,

Divakar Sharma

Academic Editor

PLOS ONE

Reviewers' comments:

Reviewer's Responses to Questions

**Comments to the Author**

1. Is the manuscript technically sound, and do the data support the conclusions?

Reviewer #1: Yes

Reviewer #2: Yes

Reviewer #3: Yes

Reviewer #4: Partly

2. Has the statistical analysis been performed appropriately and rigorously? 

Reviewer #1: Yes

Reviewer #2: Yes

Reviewer #3: Yes

Reviewer #4: Yes

3. Have the authors made all data underlying the findings in their manuscript fully available?

Reviewer #1: Yes

Reviewer #2: Yes

Reviewer #3: Yes

Reviewer #4: Yes

4. Is the manuscript presented in an intelligible fashion and written in standard English?

Reviewer #1: Yes

Reviewer #2: Yes

Reviewer #3: Yes

Reviewer #4: No

5. Review Comments to the Author

Reviewer #1: Excellent work, data clearly show the effect of temp and humidity on survival bacteria on high touch surfaces; in addition, the authors attempted to look into the underlying mechanism as well. Even though "high touched" surfaces were defined in the manuscript, "high touch" surfaces would be preferable

Reviewer #2: After reviewing the manuscript PONE-D-23-08650 titled " Human pathogenic bacteria on high-touched dry surfaces can be controlled by warming to human-skin temperature under moderate humidity, I found it very well written and clearly presented the data. I found it acceptable for publication, however, there are some points that I think they should be cleared. I have already highlighted them on the manuscript and added a sticky note stating my point. In addition, I found the quality of the figures is not of a publishable grade, especially the text on them, they should be reproduced in a better quality.

Reviewer #3: The manuscript entitled “Human pathogenic bacteria on high-touched dry surfaces can be controlled by warming to human-skin temperature under moderate humidity” investigated the dynamics of bacteria on dry surfaces in hospitals and showed experimentally that human pathogenic bacteria on dry surfaces (handrails) can be controlled by warming of the surfaces to human-skin temperature under moderate humidity. Furthermore, they demonstrated that NhaA, an Na+ /H+ antiporter, is a crucial factor in supporting the survival of bacteria under dry conditions, and that an inhibitor 2-AP of NhaA enhances the suppressive effects of warming. It contains some very interesting results, and it is suitable for publication.

In their experiment, they used a commercially available stainless steel handrail designed for doors as an experimental device. I would like to know if they have tested or intend to test with another type of material different from the one used in order to check if the result will be similar.

Reviewer #4: The authors operate from a pristigious university with a strong

research history. Collectively, they have quite interesting

publications. This manuscript is of general interest to a

broad community. Grammar, etc is good in general. The introduction

section clearly spells out a problem with references. We get

plenty of infections in the university hospital where I am and

I have since several decades been concerned about the low humidity

during our long Winters. I therefore welcome such papers and

would like to see this paper accepted.

There are some issues that make some of the most important

figures hard to follow, but I think the authors can resolve

those things. If the authors would kindly revise, I would

like to go ahead with acceptance.

1.

Abstract: Please state some actual numbers. Instead of

"significantly", state p-value.

2.

As far as I know, 70% ethanol remains among the most commonly

used cleaning agents, but is never mentioned in this manus. I

suggest the others make some mention of ethanol for completeness.

3.

Methods and Figs 3 and 4

Temperature throughout was very hard to understand. Speculating

on what makes sense to me at line 376, I think you are missing

the word "incubator". A thermo-hygrostat is only an electronic

device used to control the environment in things like incubators.

The following reflects my understanding. Hopefully it helps the

authors to understand why this was hard to follow and how to

revise.

"thermo-hygrostat incubator adjusted..."

Then, at line380, an adjacent handrail is mentioned. Maybe

something like this:

"approximately 37C when inside the incubator, as measured

with FLIR.

This now takes us back to line 377. Maybe something like

this:

"There were 2 adjacent handrails; temperatures were monitored..."

Some clarification is needed as to the differences in how

the different handrails were used.

6. PLOS authors have the option to publish the peer review history of their article (what does this mean?). If published, this will include your full peer review and any attached files.

Reviewer #1: No

Reviewer #2: No

Reviewer #3: No

Reviewer #4: No

---

## [Author Response · Author response to Decision Letter 0]

22 Aug 2023

REPLIES TO REVIEWER’S COMMENTS

Ref. ID: PONE-D-23-08650

Human pathogenic bacteria on high-touched dry surfaces can be controlled by warming to human-skin temperature under moderate humidity

To Reviewer 1: 

Comment 1: 

Excellent work, data clearly show the effect of temp and humidity on survival bacteria on high touch surfaces; in addition, the authors attempted to look into the underlying mechanism as well. Even though "high touched" surfaces were defined in the manuscript, "high touch" surfaces would be preferable. 

Response: According to the comment, we corrected it throughout the text, including the title. 

To Reviewer 2: 

Comment 1: 

After reviewing the manuscript PONE-D-23-08650 titled " Human pathogenic bacteria on high-touched dry surfaces can be controlled by warming to human-skin temperature under moderate humidity, I found it very well written and clearly presented the data. I found it acceptable for publication, however, there are some points that I think they should be cleared. I have already highlighted them on the manuscript and added a sticky note stating my point. In addition, I found the quality of the figures is not of a publishable grade, especially the text on them, they should be reproduced in a better quality.

Response: According to the comments, all corrections have been made to the points written in the text (See below), as well as improved the quality of the figures (Tiff files) with publishable grade. Specifically, the image resolution has been increased to 2,000 pixels per inch.

Responses to Highlighted points on the manuscript

Comment on line 50:

Why ineffective? some disinfectants are certainly effective. 

Response: It didn't mean that it was ineffective, but that disinfection measures alone were not enough. We apologize for the lack of explanation. According to the comment, we corrected it. 

Comment on line 196:

90% humidity is not a moderate as you said that less than 60% there was no effect for the humidity.

Response: According to the comment, we corrected it.

Comment on line 220:

Does the S. aureus also have this dry resistance factor as you found that in the E. coli library ?

Response: Yes, S. aureus is also known to show dry resistant. According to the comment, we have quoted the references in the revised text.

Comment on line 227:

In real life you might be able to control temperature but humidity is not that easy to control especially at high percentages. In general, we cant increase the humidity of hospital environments beyond a specific percentage as this might lead to other negative conditions that might surpass the benefits of decreasing the bacterial numbers on high touch surfaces. for example, might lead to the growth of mold in certain damp areas and might increase survival of desiccation resistant bacteria.

Response: We agree with the point that the reviewer pointed out. According to the comment, we added the explanation that high humidity has negative effects such as mold growth in the revised text. 

Comment on line 261:

???? is not this Clostridium. 

Response: Very sorry. As pointed out, it may not be correct to describe this bacterium as a primary pathogen of diarrhea. As pointed out by the reviewer, the text has been revised to state that diarrhea due to this bacterial infection is increasing in healthcare facilities. We hope you are satisfied with this correction.

Comment on line 312:

This is not a new method of cleaning. 

Response: According to the comment, we corrected it properly.

Comment on line 316:

Bacteria may also develop resistance to this slight increase in the temperature. Some bacteria develop resistance to dessication by acquiring genes from other bacteria. 

Response: As pointed out by the reviewer, we can't deny the possibility that, like drug-resistant bacteria, they will become resistant to human-skin warming. According to the comment, we added a note about that.

Comment on line 629:

Evidenced by live/dead staining.

Response: According to the comment, we corrected it. 

To Reviewer 3: 

Comment 1: 

The manuscript entitled “Human pathogenic bacteria on high-touched dry surfaces can be controlled by warming to human-skin temperature under moderate humidity” investigated the dynamics of bacteria on dry surfaces in hospitals and showed experimentally that human pathogenic bacteria on dry surfaces (handrails) can be controlled by warming of the surfaces to human-skin temperature under moderate humidity. Furthermore, they demonstrated that NhaA, an Na+ /H+ antiporter, is a crucial factor in supporting the survival of bacteria under dry conditions, and that an inhibitor 2-AP of NhaA enhances the suppressive effects of warming. It contains some very interesting results, and it is suitable for publication. In their experiment, they used a commercially available stainless steel handrail designed for doors as an experimental device. I would like to know if they have tested or intend to test with another type of material different from the one used in order to check if the result will be similar.

Response: Thank you for the comment. The materials used in experiments on the effect of warming to human-skin temperature under control of humidity have not yet been thoroughly verified except for polyethylene, which is the material of the well plate, and stainless steel, which is used for handrails. Meanwhile, other materials will be tested in the future. Specifically, in collaboration with handrail manufacturers, we plan to use materials that have already been installed in public environments that people touch frequently and verify them on devices capable of more accurate temperature control. Meanwhile, although it has already been published (BMC Res Notes. 2015 Dec 21;8:807. doi: 10.1186/s13104-015-1757-9), we have found that the number of surviving bacteria on high-frequency contact surfaces varies depending on the type of material. As pointed out by the reviewer, the effect of the material of the frequently touched surface with human-skin warming on bacterial survival are important issues to be verified.

To Reviewer 4:

Comment 1:

Abstract: Please state some actual numbers. Instead of "significantly", state p-value.

Response: According to the comment, statistical results have been added together with p-values in the abstract of the revised manuscript.

Comment 2:

As far as I know, 70% ethanol remains among the most commonly used cleaning agents, but is never mentioned in this manus. I suggest the others make some mention of ethanol for completeness.

Response: As pointed out by the reviewer, alcohol is effective in disinfecting high-touch surfaces. However, in order to apply the disinfection measures, it is necessary to periodically clean with alcohol, which takes time and effort. On the other hand, if frequently-touched surfaces can be kept human skin continuously, it will be possible to save time and effort in cleaning work, which may contribute to improving the quality of medical care by reducing work in busy medical settings. We added such explanation about the significance of the human-skin warming effect in the Discussion of the revised manuscript with references.

Comment 3:

Methods and Figs 3 and 4, Temperature throughout was very hard to understand. Speculating on what makes sense to me at line 376, I think you are missing the word "incubator". A thermo-hygrostat is only an electronic device used to control the environment in things like incubators. The following reflects my understanding. Hopefully it helps the authors to understand why this was hard to follow and how to revise. "thermo-hygrostat incubator adjusted..."

Response: We apologize for the lack of accuracy in the description. According to the comment, we have specified that the main experiments were conducted within a thermo-hygrostat incubator, and have reflected this in the text and figure legends.

Comment 4:

Then, at line380, an adjacent handrail is mentioned. Maybe something like this: "approximately 37C when inside the incubator, as measured with FLIR.

Response: Very sorry. That's right. We apologize for not describing it correctly. According to the comment, we corrected it.

Comment 5:

This now takes us back to line 377. Maybe something like this: "There were 2 adjacent handrails; temperatures were monitored..."

Response: Very sorry. As pointed put by the reviewer, one handrail was used as a control without heating, and the other was heated with a heater. According to the comment, we corrected it. 

Comment 6:

Some clarification is needed as to the differences in how the different handrails were used.

Response: As already mentioned, we set up two handrails inside the thermo-hygrostat incubator and conducted experiments. Also, at that time, one handrail was used as a control without a heater, and the other was heated to verify the human-skin warming effect. We have revised the description of the method to reflect this. Again, we apologize for the lack of explanations related to the experimental handrail devise with heater.

---

## [Decision Letter · Decision Letter 1]

5 Sep 2023

Human pathogenic bacteria on high-touch dry surfaces can be controlled by warming to human-skin temperature under moderate humidity

PONE-D-23-08650R1

Dear Dr. Yamaguchi

We’re pleased to inform you that your manuscript has been judged scientifically suitable for publication and will be formally accepted for publication once it meets all outstanding technical requirements.

Kind regards,

Divakar Sharma, Ph.D.

Academic Editor

PLOS ONE

Additional Editor Comments (optional):

Accept

Reviewers' comments:

Reviewer's Responses to Questions

**Comments to the Author**

1. If the authors have adequately addressed your comments raised in a previous round of review and you feel that this manuscript is now acceptable for publication, you may indicate that here to bypass the “Comments to the Author” section, enter your conflict of interest statement in the “Confidential to Editor” section, and submit your "Accept" recommendation.

Reviewer #1: All comments have been addressed

Reviewer #3: All comments have been addressed

Reviewer #4: All comments have been addressed

2. Is the manuscript technically sound, and do the data support the conclusions?

Reviewer #1: Yes

Reviewer #3: Yes

Reviewer #4: Yes

3. Has the statistical analysis been performed appropriately and rigorously? 

Reviewer #1: Yes

Reviewer #3: Yes

Reviewer #4: Yes

4. Have the authors made all data underlying the findings in their manuscript fully available?

Reviewer #1: Yes

Reviewer #3: Yes

Reviewer #4: Yes

5. Is the manuscript presented in an intelligible fashion and written in standard English?

Reviewer #1: Yes

Reviewer #3: Yes

Reviewer #4: Yes

6. Review Comments to the Author

Reviewer #1: (No Response)

Reviewer #3: The work has excellent quality and all questions were answered. Therefore, the manuscript is ready and suitable for publication.

Reviewer #4: The revision has improved the manuscript. All my concerns were addressed. Thanks for your time and good luck with publication.

7. PLOS authors have the option to publish the peer review history of their article (what does this mean?). If published, this will include your full peer review and any attached files.

Reviewer #1: No

Reviewer #3: No

Reviewer #4: No

---

## [Editor Report · Acceptance letter]

11 Sep 2023

PONE-D-23-08650R1 

Human pathogenic bacteria on high-touch dry surfaces can be controlled by warming to human-skin temperature under moderate humidity 

Dear Dr. Yamaguchi:

I'm pleased to inform you that your manuscript has been deemed suitable for publication in PLOS ONE. Congratulations! Your manuscript is now with our production department. 

Kind regards, 

on behalf of

Dr. Divakar Sharma 

Academic Editor

PLOS ONE